# Seasonal Variations and Interspecific Differences in Metabolomes of Freshwater Fish Tissues: Quantitative Metabolomic Profiles of Lenses and Gills

**DOI:** 10.3390/metabo9110264

**Published:** 2019-11-02

**Authors:** Yuri P. Tsentalovich, Vadim V. Yanshole, Lyudmila V. Yanshole, Ekaterina A. Zelentsova, Arsenty D. Melnikov, Renad Z. Sagdeev

**Affiliations:** 1International Tomography Center SB RAS, Institutskaya 3a, Novosibirsk 630090, Russia; vadim.yanshole@tomo.nsc.ru (V.V.Y.); lucy@tomo.nsc.ru (L.V.Y.); zelentsova@tomo.nsc.ru (E.A.Z.); melnikov.arsenty@tomo.nsc.ru (A.D.M.); rsagdeev@tomo.nsc.ru (R.Z.S.); 2Novosibirsk State University, Pirogova 2, Novosibirsk 630090, Russia

**Keywords:** freshwater fish, metabolomics, dissolved oxygen level, mass spectrometry, NMR spectroscopy

## Abstract

This work represents the first comprehensive report on quantitative metabolomic composition of tissues of pike-perch (*Sander lucioperca*) and Siberian roach (*Rutilus rutilus lacustris*). The total of 68 most abundant metabolites are identified and quantified in the fish lenses and gills by the combination of LC-MS and NMR. It is shown that the concentrations of some compounds in the lens are much higher than that in the gills; that indicates the importance of these metabolites for the adaptation to the specific living conditions and maintaining the homeostasis of the fish lens. The lens metabolome undergoes significant seasonal changes due to the variations of dissolved oxygen level and fish feeding activity. The most season-affected metabolites are osmolytes and antioxidants, and the most affected metabolic pathway is the histidine pathway. In late autumn, the major lens osmolytes are *N-*acetyl-histidine and threonine phosphoethanolamine (Thr-PETA), while in winter the highest concentrations were observed for serine phosphoethanolamine (Ser-PETA) and *myo*-inositol. The presence of Thr-PETA and Ser-PETA in fish tissues and their role in cell osmotic protection are reported for the first time. The obtained concentrations can be used as baseline levels for studying the influence of environmental factors on fish health.

## 1. Introduction

A complete set of small-molecular-weight compounds in a tissue—a metabolome—reflects an actual state of the tissue and may significantly vary depending on the age, diet, and health status. The role of metabolomics in the study of human and animal pathogeneses rapidly increases [1,2,3,4]. The pathologic processes in a tissue are reflected in the metabolomic changes, causing the increase or decrease of the levels of certain metabolites. The metabolomic approach has proven its efficiency in environmental studies related to aquaculture. In particular, the metabolomic analysis of fish tissues has been successfully used for the study of the impact of external factors on fish health, including water contamination with pesticides [5], herbicides [6], aromatic hydrocarbons and mercury [7,8], low oxygen level [9], water temperature [10,11]. A recent review [12] is devoted to the application of the metabolomic approach to the study of viral, bacterial, and parasite fish diseases.

One can assume that metabolically active tissues—gills, blood, liver, meat—are more sensitive to environmental factors than relatively conservative tissues such as bones, lens, or vitreous humor. For that reason, the metabolically active fish tissues are often used for the analysis. On the other hand, the conservative tissues may accumulate the metabolomic changes induced by adverse ecological factors or by diseases, and, therefore, give additional information on the mechanism of disease development. The metabolomic study of the fish lens attracts a special interest in the view of reports on extremely high cataract prevalence in farmed fish (especially salmonids) [13,14,15], in some cases reaching 100%. It has been suggested [11,16,17,18] that the disease cause is the low supply of amino acid histidine, used for the biosynthesis of the major osmolyte of the fish lens, *N-*acetyl-histidine (NAH) [19,20]. Indeed, low histidine level in the fish feed results in the low concentration of NAH in the eye lens; insufficient tissue buffering and osmoregulation causes the development of osmotic cataracts, especially after the seawater transfer [17].

The eye lens mostly consists of fiber cells without nuclei and organelles lost during the cell differentiation [21]. That makes the lens transparent in the visible light range, but the metabolic activity inside the lens is minimal. Therefore, the lens defense almost completely relies on small molecules either entering the lens from the surrounding aqueous humor (AH), or synthesized in metabolically active lens epithelial monolayer. First of all, these molecules include antioxidants preventing the development of oxidative stress, and osmolytes maintaining the intracellular pressure. Lack of antioxidants or osmoprotectants may lead to the damage of eye tissues, and to cataract development. The proper lens functioning also requires the constant supply of nutrients for cellular energy generation, biochemical synthesis, and other cellular functions.

It is important to notice that the majority of published metabolomic data are semi-quantitative, yielding the difference between the metabolomic profiles of experimental and control samples. The quantitative data on the metabolomic composition of fish tissues (metabolite concentrations in moles per gram of tissue) can be found only in a few papers for rather limited set of compounds. Recently, we developed an approach based on the combined application of NMR spectroscopy and liquid chromatography with mass-spectrometric and optical detection (LC-MS) to the quantitative metabolomic profiling of biological liquids and tissues [22,23,24,25,26,27]. This approach allows for the determining of concentrations of up to one hundred compounds in a sample. In the present paper, we apply this approach for the metabolomic profiling of lenses and gills of two kinds of freshwater fish—pike-perch (*Sander lucioperca*) and Siberian roach (*Rutilus rutilus lacustris*), inhabiting the basin of the Siberian river Ob. The major goals of the work are:(a)To determine the major metabolites present in the fish lens and gill, including osmolytes, antioxidants, amino acids, organic acids etc., and to measure their concentrations;(b)To compare the metabolomic profiles of the fish lens and gill. The gill is a very blood-rich tissue, while the metabolomic composition of AH surrounding the lens is very similar to that of blood plasma [22]. Therefore, the comparison of metabolomic compositions of the lens and gill may help to determine which compounds enter the lens from blood via AH, and which ones are specifically synthesized inside the lens;(c)To compare the metabolomic composition of gills and lenses from herbivorous–omnivorous (*R. rutilus lacustris*) and predatory (*S. lucioperca*) fish;(d)To compare the lens metabolomic composition of fish caught at different times of year in order to estimate the influence of a seasonal factor on the lens metabolomic profile.

## 2. Results

### 2.1. Metabolite Identification

Figure 1 and Figure 2 show the NMR spectra of protein-free lipid-free extracts from the *S. lucioperca* lens and gill. Most of the signals in the NMR spectra correspond to well-known metabolites whose spectra are available in literature [6,7,8,9,24,26,28,29,30]. These metabolites include amino acids, organic acids, alcohols, sugars, nucleotides, and others. For the majority of these compounds, the identification was performed according to their NMR spectra without additional confirmation. In some cases, the signal assignment was unobvious; in these cases, the identification was confirmed by spiking the extract with commercial standard compounds.

After preliminary metabolite identification, few major signals in the NMR spectra remained unassigned, including doublet at 1.45 ppm and 3 multiplets at 4.23 ppm, 4.30 ppm, 4.82 ppm. To assign these signals, the lens extract was chromatographically separated into 15 fractions, and each fraction was subjected to MS, MS/MS, and NMR analysis. That made it possible to identify two unknown metabolites, namely threonine phosphoethanolamine (Thr-PETA) and serine phosphoethanolamine (Ser-PETA). Quantitative analysis shows (see below) that these compounds are among the most abundant metabolites in the fish lens and gill. The chemical structures of these compounds are shown in Scheme 1, and their MS/MS and NMR properties are listed below.

Thr-PETA:

^1^H NMR (700 MHz, D_2_O): 1.452 (3H, d, J = 6.7 Hz, CH_3_); 3.247 (2H, m, CH_2_); 3.736 (1H, dd, J = 2.2, 2.5 Hz, CH); 4.039 (2H, q, J = 5.3 Hz, CH_2_); 4.820 (1H, m, CH).

MS/MS (ESI^+^, 20.7 eV): 243.0737 [C_6_H_16_N_2_O_6_P^+^], 200.0321 [C_4_H_11_NO_6_P^+^], 142.0269 [C_2_H_9_NOP^+^], 120.0658 [C_4_H_10_NO_3_^+^], 102.0553 [C_4_H_8_NO_2_^+^], 98.9839 [H_4_O_4_P^+^], 84.0446 [C_4_H_6_NO^+^].

Ser-PETA:

^1^H NMR (700 MHz, D_2_O): 3.267 (2H, m, CH_2_); 3.978 (1H, m, CH); 4.079 (2H, m, CH_2_); 4.23 (1H, m, CH_2_); 4.29 (1H, m, CH_2_).

MS/MS (ESI^+^, 20.5 eV): 229.0586 [C_5_H_14_N_2_O_6_P^+^], 186.0166 [C_3_H_9_NO_6_P^+^], 166.0166 [C_4_H_9_NO_4_P^+^], 142.0267 [C_2_H_9_NO_4_P^+^], 106.0499 [C_3_H_8_NO_3_^+^], 98.9843 [H_4_O_4_P^+^], 88.0396 [C_3_H_6_NO_2_^+^].

More detailed data on NMR and MS/MS characterization of Thr-PETA and Ser-PETA are presented in Appendix A.

The annotation of signals in LC-MS spectra was performed by the search in online databases (HMDB, METLIN, ChemSpider, ChEBI) using the obtained exact m/z values and verified by the analysis on isotopic pattern, retention time, characteristic fragment or adduct ions, and MS/MS spectra. The compounds identified by LC-MS were checked by NMR data, and vice versa. For 26 compounds to be further quantified by LC-MS, solid verification (MSI guidelines level 1) was done by the injection of chemical standard samples. In the present work, only the metabolites quantified by NMR or LC-MS or by both methods were considered.

### 2.2. Metabolite Quantification

The measurements of metabolite concentrations were performed for tissues of *S. lucioperca* and *R. rutilus lacustris* caught in the Ob reservoir (Novosibirsk region, Siberia, Russia) during the late autumn (October–November) and winter (February) periods. Typically, ice freezes in Novosibirsk region in the second half of November, while ice-breaking occurs at the end of April. It is known that the level of dissolved oxygen (DO) in ice-covered lakes is maximal during the late autumn (before ice freezing), and minimal in winter due to the water isolation from atmosphere [31]. Therefore, the data on autumn fish correspond to high DO season, and on winter fish, to low DO season.

The concentrations of metabolites in lenses and gills were measured by two methods. The first method includes the integration of NMR signals in the spectra of tissue extracts relatively to the internal standard DSS followed by the recalculation of the metabolite concentration in the sample to the metabolite concentration in the tissue (in nmoles per gram of the tissue wet weight). The LC-MS-based quantification was performed using the external calibration curves constructed for each metabolite under study. The majority of metabolites were quantified using the NMR method, while LC-MS method was mostly used for metabolites whose NMR signals were strongly overlapped by signals of other compounds, or the concentrations were too low for the reliable NMR quantification. For additional control, the concentrations of several metabolites were measured by both NMR and LC-MS methods. The concentrations of a total of 68 metabolites present in the fish lens and gill have been determined. The results of the measurements are presented in Table 1, more detailed information can be found in Appendix A. Table 1 also indicates which method was used for metabolite quantification, NMR or LC-MS. A symbol NMR* corresponds to metabolites, whose concentrations were measured by NMR and confirmed by LC-MS (both methods gave similar results). The metabolites in Table 1 are divided into six groups according to their chemical and biological properties and functionalities: “Amino acids”, “Organic acids”, “Alcohols, amines, and sugars”, ”Osmolytes”, “Antioxidants”, and “Nitrogenous bases, nucleotides, nucleosides”. 

### 2.3. Quantitative Data Analysis

To get the overview of the general metabolomic differences between the four lens groups, the data on metabolite concentrations in lenses were subjected to a principal component analysis (PCA). The following groups were analyzed: lenses from *S. lucioperca* caught in autumn and in winter, and lenses from *R. rutilus lacustris* caught in autumn and in winter. Figure 3 (left panel) shows the PCA scores plot for the 1st principal component (PC1 = 45.1% of explained variance) versus the 2nd component (PC2 = 25.8% of explained variance). The corresponding loadings plot is presented in Figure 3 (right panel).

The data in the PCA scores plot are concentrated into four distinct groups, which demonstrates that: (a) the metabolomic compositions of *S. lucioperca* and *R. rutilus lacustris* lenses differ significantly, and the genera are separated along the PC1 axis; (b) the metabolomic profiles of lenses from both fish genera undergo noticeable seasonal variations spread along the PC2 axis; (c) within each experimental group, the data scattering is rather small.

Figure 4 shows boxplots for concentrations of nine metabolites which demonstrate the most significant differences between genera and the most pronounced seasonal variations. *myo*-Inositol, NAH, Thr-PETA are the most discriminative metabolites between genera, and they demonstrate significant seasonal dependences (Figure 3). Large seasonal variations are also observed for ADP, histidine, *N-*acetyl-aspartate (NAA), OSH, serine, and Ser-PETA.

The differences in the metabolite concentrations in the fish lens and gill were calculated as the ratio of averaged concentrations in the lens to that in the gill (lens/gill ratio). To reveal statistically important differences between the groups, the Mann–Whitney *U*-test (with the use of FDR correction) was performed. The resulting barplots containing graphical information on the lens/gill ratios are presented in Figure 5 (left panel for *S. lucioperca* and right panel for *R. rutilus lacustris*). Only the metabolites which differ significantly (*p* < 0.05, fold change > 1.5) are shown. Numerical values can be found in Appendix A. The bars expanding to the left from the unity correspond to the elevated level of a metabolite in the gill, and to the right, to the elevated level in the lens. For several metabolites the ratio of their levels in lens to that in the gill (or vice versa) exceeds two orders of magnitude. For the majority of such values that means that a metabolite was not detected in either gill or lens, correspondingly. In fact, the numerical ratio of two significantly different values is rather unreliable; therefore we decided to cut all bars at the level of the lens/gill and gill/lens ratios equal to 30. The lenses of both fish types contain elevated levels of NAH, *N-*acetyl-3-methyl-histidine (*N-*Ac-3-Me-His), NAA, NAD, and tryptophan, while the reduced concentrations (as compared to gills) are observed for xanthine, uracil, hypoxanthine, guanosine, ethanolamine (ETA), isobutyrate, γ-aminobutyrate (GABA), and inosine.

For further investigation of pathways involved in the seasonal variations occurring in the fish lens, we performed quantitative metabolite set enrichment analysis (MSEA), comparing the metabolite concentrations in autumn and in winter. The assignment of metabolites to particular pathways was performed with the use of MetaboAnalyst web platform [32] using self-defined metabolite sets based on the SMPDB (Small Molecule Pathway Database) library as described in the Materials and Methods section. The results of MSEA are presented in Figure 6 (left panel for *S. lucioperca* and right panel for *R. rutilus lacustris*) and in Appendix A. The “Histidine metabolism” pathway was modified by addition of NAH, OSH, and their intermediate compounds to the list of metabolites already existing in the SMPDB library; an asterisk sign (*) indicates this modification. The “Histidine metabolism*” is the most season-affected pathway for both genera. The following pathways are also amongst the most affected in both genera: “Citric acid cycle”, “Arginine and proline metabolism”. Besides, in *S. lucioperca* pronounced changes are observed in “Ammonia recycling”, “Galactose metabolism”, and “Inositol metabolism”. For *R. rutilus lacustris*, the “Phospholipid biosynthesis”, “Phenylalanine and tyrosine metabolism”, and “Cysteine metabolism” are amongst the most affected pathways.

### 2.4. Metabolite Group Analysis

The information presented in the Table 1 and Figure 3, Figure 4, Figure 5 and Figure 6 was used for the analysis of the metabolite concentrations in the lenses and gills of *S. lucioperca* and *R. rutilus lacustris*. The main objectives of the analysis were to establish the most abundant metabolites in tissues and to reveal similarities and differences between sample groups.

#### 2.4.1. Amino Acids

Alanine and glutamine are the most abundant amino acids (with the exception of taurine which was placed in the group “Osmolytes”) in lenses and gills of both *S. lucioperca* and *R. rutilus lacustris*.

The amino acid compositions of gills of *S. lucioperca* and *R. rutilus lacustris* are rather similar. Noticeable differences (by a factor of ca. 3) were observed for glutamine, lysine, serine, sarcosine, creatine, carnitine, and *N-*acetyl-carnitine; for all these amino acids, their concentrations in the *R. rutilus lacustris* gills were higher. The only amino acids prevailing in the *S. lucioperca* gills are tryptophan and betaine.

The levels of the majority of amino acids in lenses of *S. lucioperca* and *R. rutilus lacustris* are also similar. The concentrations of valine, isoleucine, methionine, asparagine, phenylalanine, *N-*Ac-3-Met-His, and sarcosine are higher in the *S. lucioperca* lens, while the concentrations of serine, carnitine, and betaine—in the *R. rutilus lacustris* lens.

The comparison of amino acid compositions of the fish gills and lens (Figure 5) shows that the lens contains elevated levels of leucine, methionine, glutamine, tyrosine, histidine, phenylalanine, and tryptophan. The concentration of the latter in lenses is higher than that in the gills by an order of magnitude. The reduced concentrations in lenses as compared to gills were observed for lysine, glycine, proline, sarcosine, creatine, carnitine, and betaine.

#### 2.4.2. Organic Acids

The organic acids listed in Table 1 are mostly intermediates or final products of metabolic reactions. Since the metabolism in the gill is much more active than that in the lens, the concentrations of the majority of organic acids in the gill are higher than in the lens. Especially high gill/lens ratios were found for GABA and fumarate. The prevalence in the lens was observed only for two acids, 2-hydroxy-butyrate and α-aminobutyrate (AABA). Lactate, being the final glycolysis product, is by far the most abundant acid in the fish lens and gills. For the majority of organic acids, their levels in the lens in autumn are higher than in winter.

#### 2.4.3. Alcohols, Amines, and Sugars

Similarly to organic acids, the levels of the majority of metabolites from this group in gills are much higher than that in lenses, with the only exception of phosphocholine. The lowest levels in lenses as compared to gills were observed for ethanolamine, choline, and glycerol.

#### 2.4.4. Osmolytes

The most abundant compound in the *S. lucioperca* gills is *myo*-inositol (8.2 µmol/g) followed by taurine (5.5 µmol/g) and Ser-PETA (3.3 µmol/g). Apparently, one of the major roles of these metabolites in the fish blood and gill tissue is the cellular osmotic protection. Thr-PETA (1.6 µmol/g) most likely also participates in the osmotic protection. The same compounds play the role of osmoprotectants also in the *R. rutilus lacustris* gills, but their abundances differ: The most abundant metabolite is taurine (8.7 µmol/g) followed by Ser-PETA (3.8 µmol/g), *myo*-inositol (2.0 µmol/g), and Thr-PETA (1.4 µmol/g).

The major osmolytes in the fish lens differ significantly from that in the fish gill. Firstly, the concentrations of taurine in lenses (0.2–0.5 µmol/g) are much smaller than in gills. Secondly, the lens contains high concentrations of *N-*acetyl-histidine (NAH) and *N-*acetyl-aspartate (NAA). Thus, the full list of the major osmolytes in the lens of both *S. lucioperca* and *S. lucioperca* includes *myo*-inositol, Thr-PETA, Ser-PETA, NAH, and NAA. However, the abundances of these metabolites in lenses of *S. lucioperca* and *R. rutilus lacustris* differ, and their levels undergo significant seasonal changes. In lenses of autumn *S. lucioperca*, the most abundant osmolytes are NAH and Thr-PETA followed by NAA, Ser-PETA, and *myo*-inositol. During the winter, the levels of NAH, Thr-PETA, and NAA decrease, while the concentrations of Ser-PETA and *myo*-inositol increase. As a result, in winter, Ser-PETA and *myo*-inositol become the most abundant osmolytes in the *S. lucioperca* lens (Table 1). Similar seasonal changes occur in the lens of *R. rutilus lacustris*: NAH, Ser-PETA, and Thr-PETA prevail in autumn, while Ser-PETA and *myo*-inositol—in winter.

#### 2.4.5. Antioxidants

In gills of *R. rutilus lacustris*, three major antioxidants—ovothiol A (1-methyl-4-thiol-L-histidine, OSH), glutathione (GSH), and ascorbate—are present in similar concentrations of approximately 100 nmol/g. In gills of *S. lucioperca*, the level of OSH is two times higher, while the levels of GSH and ascorbate are significantly lower. Ergothioneine was detected (but not quantified) only in gills of *R. rutilus lacustris*.

The concentration of OSH in lenses of both *S. lucioperca* and *R. rutilus lacustris* is significantly higher than in gills. Especially high levels of OSH were detected in lenses of autumn fish—3 µmol/g for *S. lucioperca* and 1.1 µmol/g for *R. rutilus lacustris*. In winter, the levels of OSH in lenses significantly drop (two-fold for *S. lucioperca* and four-fold for *R. rutilus lacustris*). The seasonal variations of GSH in the lens are less pronounced: practically no changes were found for *S. lucioperca*, and approximately a two-fold decrease in winter for *R. rutilus lacustris*. The level of ascorbate varies in the fish lens from 30 to 100 nmol/g, ergothioneine in lenses was not detected.

#### 2.4.6. Nitrogenous Bases, Nucleotides, Nucleosides

Most of compounds in this group are the products of intracellular biosynthesis, so one can expect that their concentrations in a tissue depend on the metabolic activity. Indeed, lens/gill ratio >1 was found only for ATP, ADP, AMP, and NAD. No significant differences were found between tissues of *S. lucioperca* and *R. rutilus lacustris*, the seasonal variations of the lenticular levels for the majority of metabolites from this group are also minimal.

## 3. Discussion

The present work is the first report on the detailed quantitative metabolomic composition of the fish tissues. Measurements were performed for lenses and gills from two types of freshwater fish—*S. lucioperca* and *R. rutilus lacustris*. The gill is a metabolically active blood-rich tissue, while the metabolic activity inside the lens is minimal. Although the metabolome is implicitly transient, the intracellular metabolites can stay inside the anatomically isolated tissue (such as the eye lens) for a long time. Thus, one can expect that the metabolomic composition of the gills reacts to the external factors promptly but reversibly, while the metabolomic changes in the lens accumulate with time. The obtained data were used for the comparison of metabolomic profiles of the lens and gill belonging to the same fish; study of seasonal variations of the lens metabolomic composition; the comparison of the metabolomic composition of tissues belonging to herbivorous–omnivorous (*R. rutilus lacustris*) and predatory (*S. lucioperca*) fishes.

The comparison of metabolomic profiles of fish gills shows that the levels of the majority of amino acids in *R. rutilus lacustris* gills are higher than that in *S. lucioperca* gills. *R. rutilus lacustris* gills also contain higher concentrations of AABA, GABA, glucose, ascorbate, GSH, AMP, and NAD. Most likely, the observed difference should be attributed to the different types of food of omnivorous and predatory fishes. In lenses, one can see a different picture: The concentrations of many metabolites in the *S. lucioperca* lens are significantly higher than in the *R. rutilus lacustris* lens. These metabolites include amino acids (isoleucine, methionine, sarcosine, valine), osmolytes (NAA), antioxidants (GSH and OSH). The amino acids are ingested with food, while NAA, GSH, and OSH are the products of intracellular biosynthesis. Therefore, it is doubtful whether the difference in the lens metabolomic compositions of *S. lucioperca* and *R. rutilus lacustris* corresponds to the different types of food only—more likely, the difference also originates from the different contributions of metabolic pathways formed during the evolution of two fish genera.

There are a significant number of metabolites whose concentrations in the gills are rather high, while their lenticular levels are either low or were not detected at all. The most significant difference in concentrations between gills and lens was found for GABA, ETA, isobutyrate, taurine, and a group of nitrogenous bases and nucleosides: inosine, guanosine, xanthine, hypoxanthine, uracil, uridine (Figure 5). It is known that the metabolomic composition of the blood plasma is very similar to that of aqueous humor surrounding the lens and providing the lens nutrition and waste removal [22]. Therefore, the majority of these compounds should be attributed to the intracellular metabolites of the gills tissue and blood. In the lens cells the metabolic activity is low, and these compounds are either not produced or produced in much lower amounts.

On the other hand, there are lens-specific compounds whose levels in the lens are significantly higher than in the gill. These metabolites include lenticular osmolytes NAH and NAA, and antioxidants OSH and GSH. Most likely, these compounds are synthesized in metabolically-active lens epithelial cells for the lens protection against osmotic and oxidative stresses. Surprisingly, the lens also contains an enhanced level of ATP despite the low metabolic activity in the lens cells. It can be assumed that the high lenticular ATP level corresponds to either activity of Na^+^/K^+^ pumps governing the water circulation through the lens [33,34,35] or to the mitochondrial activity taking place in the lens epithelial layer and outer cortex, and providing very low oxygen concentration in the inner parts of the lens [36].

There are three major seasonal factors which can affect the metabolomic composition of the fish tissues: the water temperature, the DO level, and the fish feeding activity. In the present study, the temperature factor is excluded: The water temperatures in late autumn and winter are similar, 4–7 °C. Low DO level during the winter leads to the deceleration of metabolic processes in fish. The metabolomic composition of the fish lens undergoes significant seasonal changes due to the low DO level and low feeding activity in winter. The most pronounced decrease of the lenticular level in winter was observed for amino acids creatine, glutamine, histidine, and *N-*Ac-3-Me-His; for the majority of organic acids; for osmolytes NAH and Thr-PETA; for antioxidant OSH. The winter decrease of organic acid levels in the fish lens probably reflects the deceleration of metabolic reactions, and, correspondingly, the lower rate of the generation of metabolic products.

To the best of our knowledge, till now NAH was the only well-recognized osmolyte in the fish lens [19,20]. In the present work, we have shown that several compounds protect the lens cells from the osmotic stress: *myo*-inositol, Thr-PETA, Ser-PETA, NAH, and NAA. In this regard, the fish lens significantly differs from mammalian lenses: For example, the major osmolytes in the rat lens are taurine and hypotaurine [25], and in the human lens—*myo*-inositol [24,30]. The list of osmolytes in the fish lens also differs from that in the fish gill, where high level of taurine was observed, while NAH is present in rather low concentration. The complex composition of osmolytes in the fish lens is probably an evolutionary response to the seasonal variations of the environment: During the periods of low oxygen content in the water and low feeding activity, the histidine supply may become insufficient for NAH synthesis, and the lens osmotic protection relies on other metabolites, such as *myo*-inositol and Ser-PETA. In this work, the presence of Thr-PETA and Ser-PETA in the fish tissues and their role in the cell osmotic protection are reported for the first time.

In the vast majority of animal tissues, GSH, ergothioneine, and ascorbate are the main antioxidants providing the deactivation of free radicals and the reduction of oxidized molecules [37,38,39]. Our lab has recently reported the finding of high concentrations of OSH in the fish lens [26]. The results of this work confirm that the major antioxidant of the fish lens is OSH. OSH is one of the strongest antioxidants existing in nature: Since pKa value of the thiol group is very low (pKa ≈ 1.0–1.4 [40,41,42,43], under physiological conditions OSH exists predominantly in highly reactive thiolate form. For that reason, the oxidation potential of OSH is significantly lower than that of GSH [44], and the oxidation of OSH by electron acceptors proceeds with the higher rate constants than that for GSH [40,45]. Earlier, OSH and its methylated derivatives were found in eggs and ovarian tissue of marine invertebrates (such as sea urchin, sea star, scallop, octopus) [46,47,48,49]. It was supposed [26] that in the fish lens, OSH represents the first line of the cellular defense against oxidative stress, reducing reactive oxygen species. Oxidized ovothiol molecules OSSO are then reduced by GSH [50], and oxidized glutathione GSSG is reduced by glutathione reductase. This reaction scheme might explain high concentrations of NAD in the fish lens, since the enzymatic reduction of GSSG requires the participation of NAD(P)H in the reaction.

OSH was also detected in the fish gills. Its level in the gill is much lower than in the lens; nevertheless, the concentration of OSH in the gill is similar or higher than the concentration of GSH. Therefore, one can assume that OSH plays an important role as an intracellular antioxidant not only in the lens, but also in other fish tissues.

The results of this work point to the importance of histidine supply in fish food. This amino acid is used for biosynthesis of two metabolites playing a vital role in maintaining homeostasis in the fish lens—the main lens osmolyte NAH and the main lens antioxidant OSH. Low feeding activity and deceleration of metabolic processes in winter cause a drop in lenticular levels of histidine, NAH, and OSH. The NAH deficiency can be compensated by synthesis of other osmolytes (*myo*-inositol and Ser-PETA), but the lack of OSH makes the lens tissue significantly more vulnerable to the oxidative stress. In particular, several publications [10,13,15,16,18] reported that the low dietary histidine supplementation provokes the cataractogenesis in farmed salmon, which was attributed to the decrease of the NAH level in the lens and the development of osmotic cataract. It is possible that the lack of OSH in the lens also makes a significant contribution to cataract development.

## 4. Materials and Methods

### 4.1. Chemicals

Chloroform, methanol, and acetonitrile (HPLC grade) were purchased from Panreac (Barcelona, Spain). D_2_O 99.9% was purchased from Armar Chemicals (Dottingen, Switzerland). All other chemicals were purchased from Sigma-Aldrich (St. Louis, MO, USA). H_2_O was deionized using Ultra Clear UV plus TM water system (SG water, Hamburg, Germany) to the quality of 18.2 MOhm.

### 4.2. Fish Sample Collection

The study was conducted in accordance with the ARVO Statement for the Use of Animals in Ophthalmic and Vision Research and the European Union Directive 2010/63/EU on the protection of animals used for scientific purposes, and with the ethical approval from the International Tomography Center (ECITC-2017-02). No special permission from the national or local authorities is required.

Pike-perch (*Sander lucioperca*, body weight 200–300 g) and Siberian roach (*Rutilus rutilus lacustris*, body weight 80–110 g) were caught in the Ob reservoir: *S. lucioperca*—in October (*n* = 8) and February (*n* = 7); *R. rutilus lacustris*—in November (*n* = 10) and February (*n* = 8). The exact dates and conditions of catching are given in Appendix A. The fish were killed with a concussive blow to the head immediately after the catching, the lenses and gills were cut from the fish, frozen and kept at −70 °C until analyzed.

### 4.3. Fish Lens and Gill Preparation

Each fish lens was weighed prior to homogenization: for *S. lucioperca*, the typical lens weight was 100 mg, and for *R. rutilus lacustris*—40 mg. Only one lens from each fish was used for the analysis. The lens was placed in a glass vial and homogenized with a TissueRuptor II homogenizer (Qiagen, Netherlands) in 1600 µL of cold (−20 °C) MeOH, and then, 800 µL of water and 1600 µL of cold chloroform were added. The mixture was shaken well in a shaker for 20 min and left at −20 °C for 30 min. Then the mixture was centrifuged at 16,100× *g*, +4 °C for 30 min, yielding two immiscible liquid layers separated by a protein layer. The upper aqueous layer (MeOH-H_2_O) was collected, divided into two parts for NMR (2/3) and LC-MS (1/3) analyses, and lyophilized.

Each fish gill was divided into arch and filaments. Only gill filaments were used for the analysis. Samples were weighed prior to homogenization: for *S. lucioperca*, the typical gill filament weight was 95 mg, and for *R. rutilus lacustris*—110 mg. The homogenization and extraction procedures for gill filaments were performed in the same way as for fish lenses.

### 4.4. NMR Measurements

The extracts for NMR measurements were re-dissolved in 600 μL of D_2_O containing 6 × 10^−6^ M sodium 4,4-dimethyl-4-silapentane-1-sulfonic acid (DSS) as an internal standard and 20 mM deuterated phosphate buffer to maintain pH 7.2.

The ^1^H NMR measurements were carried out at the Center of Collective Use «Mass spectrometric investigations» SB RAS on a NMR spectrometer AVANCE III HD 700 MHz (Bruker BioSpin, Rheinstetten, Germany) equipped with a 16.44 Tesla Ascend cryomagnet. The proton NMR spectra for each sample were obtained with 96 accumulations. Temperature of the sample during the data acquisition was kept at 25 °C, the detection pulse was 90 degree. The repetition time between scans was 20 s to allow for the relaxation of all spins. Low power radiation at the water resonance frequency was applied prior to acquisition to presaturate the water signal. The concentrations of metabolites in the samples were determined by the peak area integration respectively to the internal standard DSS.

### 4.5. LC-MS Measurements

The extracts for LC-MS analysis were re-dissolved in 100 μL of aqueous solution containing 10 μM *N-*acetyltryptophanamide as an internal control. For each sample, three dilutions (1, 1/4, 1/16) were made to extend the coverage of metabolite concentrations.

The LC separation was performed on a UltiMate 3000RS chromatograph (Dionex, Germering, Germany) using a hydrophilic interaction liquid chromatography (HILIC) method on a TSKgel Amide-80 HR (Tosoh Bioscience, Griesheim, Germany) column (4.6 × 250 mm, 5 μm) as described earlier [26]. The chromatograph was equipped with a flow cell diode array UV-vis detector (DAD) with 190–800 nm spectral range. Solvent A consisted of 0.1% formic acid solution in H_2_O, solvent B consisted of 0.1% formic acid solution in acetonitrile. The gradient was (solvent B): 95% (0–5 min), 95%–65% (5–32 min), 65%–35% (32–40 min), 35% (40–48 min), 35%–95% (48–50 min), 95% (50–60 min); the flow rate was 1 mL/min, the sample injection volume was 10 L. After the DAD cell, a home-made flow splitter (1:10) directed the lesser flow to an ESI-q-TOF high-resolution hybrid mass spectrometer maXis 4G (Bruker Daltonics, Bremen, Germany). The mass spectra were recorded in a positive mode with 50–1000 m/z range. The MS setup, the calibration procedure, and the data processing were described in detail earlier [25,26,27,51]. Briefly, eight solutions containing an equimolar mixture of 26 metabolites with the concentrations ranging from 43.5 to 87 μM were subjected to LC-MS, and the calibration curves for each metabolite were plotted.

### 4.6. LC Fraction Collection

To get the overview of the general metabolomic differences between the four lens groups, the data on metabolite concentrations in lenses were subjected to a principal component analysis (PCA). The following groups were analyzed: lenses from *S. lucioperca* caught in autumn and in winter, and lenses from *R. rutilus lacustris* caught in autumn and in winter. Figure 3 (left panel) shows the PCA scores plot for the 1st principal component (PC1 = 45.1% of explained variance) versus 2nd component (PC2 = 25.8% of explained variance). The corresponding loadings plot is presented in Figure 3 (right panel).

### 4.7. Data Analysis

MS/MS spectra of unknown compounds were analyzed with the MetFrag web tool [52] using ChemSpider and PubChem databases. The tool was used for the retrieving compounds from databases according to the measured exact mass and sorting compounds according to the score for the most possible candidates based on exact mass of fragments in MS/MS spectra.

To explore the data and to display the general metabolomic dependences in the data, the principal component analysis (PCA) was performed on a MetaboAnalyst 4.0 web-platform (www.metaboanalyst.ca [32]). PCA scores and loadings plots were constructed with the range data scaling to normalize the contributions of all metabolites.

To reveal the biochemical pathways that are mostly affected by the seasonal variations in the fish lens, we performed the metabolite set enrichment analysis (MSEA) on the MetaboAnalyst platform. MSEA was performed for quantitative data without prior normalization. For the analysis, self-defined metabolite sets were used: 27 metabolic cycles were selected from the 99 pathway-associated metabolite sets from the Small Molecules Pathways Database (www.smpdb.ca) containing at least one of the quantified metabolite. The rest of the pathways were excluded from the MSEA, as they contained ubiquitous non-informative metabolites (e.g., ATP, ADP, NAD) or did not contain any of the quantified metabolites. The histidine metabolic pathway was extended by the addition of NAH, ovothiol A (OSH), and their precursors described earlier in literature [53,54].

To reveal the statistically-important differences between the groups (lens/gill, seasons), the Mann–Whitney *U*-test was performed using *scipy* (v1.1.0) module on *Python*. The correction of *p*-values for the multiple comparisons by the false discovery rate (FDR) method was done according to the Benjamini–Hochberg procedure using *statsmodels* (v0.9.0) module for *Python*.

## 5. Conclusions

At the moment, the detailed quantitative metabolomic composition is known for very limited number of tissues and species (except human tissues). One of the few examples of quantitative metabolomic analysis of marine animals is a recent paper by Cappello et al. [28] devoted to the analysis of mussel tissues. In the present work, the concentrations of a broad spectrum of metabolites (68 compounds) in the fish lens and gills are measured for the first time. The obtained quantitative data can be used as baseline levels of metabolites for studying the influence of environmental factors (such as water temperature, DO level, water pollution, diet) on fish health. It is found that the metabolomic composition of the fish lens undergoes strong seasonal variations caused by changes in the DO level, fish feeding activity, and probably other factors. In metabolically passive lens fiber cells, the intracellular defense mostly relies on metabolites—osmolytes and antioxidants. The major lens antioxidant is OSH, while the osmotic protection is provided by the combination of *myo*-inositol, Thr-PETA, Ser-PETA, NAH, and NAA. The concentrations of these compounds and their roles in cytoprotection vary with season: In particular, in the late autumn, NAH and Thr-PETA are the main lens osmolytes, while in February, Ser-PETA and *myo*-inositol become the most abundant osmolytes. In the fish gills, three antioxidants—OSH, GSH, and ascorbate—are present in similar concentrations, and the main osmolytes are *myo*-inositol, taurine, and Ser-PETA. It is important to notice that the main lenticular antioxidant OSH and one of the major lenticular osmolytes NAH are synthesized from the same precursor, amino acid histidine. That indicates the importance of the histidine supply in the fish diet for maintaining homeostasis in the fish lens. The present study was performed for freshwater fish. It would be interesting to compare the metabolomic compositions of tissues of freshwater and marine fish; this work is currently in progress in our laboratory.

## Data Availability

The data obtained in this study including NMR raw data, metabolite concentrations, and experimental protocols have been deposited in MetaboLights repository, study identifier MTBLS1057 (https://www.ebi.ac.uk/metabolights/MTBLS1057).

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
