# Peer review of "Seasonal Variations and Interspecific Differences in Metabolomes of Freshwater Fish Tissues: Quantitative Metabolomic Profiles of Lenses and Gills"

_metabolites, 2019, doi:10.3390/metabo9110264_

Round 1

Reviewer 1 Report

The manuscript presents a quantitative metabolomic study of lens and gill from freshwater fish. The authors compare interspecific and seasonal variability of munerous compounds characterized in these tissues related to their metabolic pathway.

In my opinion, this work is relevant and the experimental design is appropriate to achieve the objectives mentioned. The combination of both proton NMR and mass specrtoscopy allowed the authors to extend the number of identified and quantified metabolites. They also applied their mastery of this approach, which they previously used to quantify rat lens metabolites. I am convinced by the structures proposed for the new compounds but I would have liked confirmation by 2D NMR data.

However, I have some remarks or questions :

Line 74: “lens and gills » singular or plural ?

Line 131: I would have said putative “annotation”

Line133: How did the authors compare LC-MS retention times of the unknown compounds with data reported in databases or in literature ?

Lines 135-136: “For 26 compounds to be further quantified by LC-MS, solid verification (MSI guidelines level 1) was done by the injection of chemical standard samples ». Which ones ? Perhaps the authors could differentiate those one from the putative identified compounds.

Line 191-192: « These conclusions are in a good agreement with the quantitative metabolomic data present in Table 1. » In my opinion this sentense should be removed because data from table 1 have been used to build PCA score plots.

In conclusion, I suggest to the editor to accept this paper after minor revision.

Author Response

Comment: Line 74: “lens and gills » singular or plural ?

Reply: thanks, corrected

Comment: Line 131: I would have said putative “annotation”

Reply: It is already written “annotation” in the text. In our opinion, is a softer term than “assignment”, and it implies some putative character.

Comment: Line133: How did the authors compare LC-MS retention times of the unknown compounds with data reported in databases or in literature?

Reply: The retention times were used only for compounds with the known RTs. However, in most cases the data on exact m/z value, isotopic pattern, characteristic fragment or adduct ions, and MS/MS spectra are sufficient for the reliable annotation of LC-MS signals.

Comment: Lines 135-136: “For 26 compounds to be further quantified by LC-MS, solid verification (MSI guidelines level 1) was done by the injection of chemical standard samples ». Which ones ? Perhaps the authors could differentiate those one from the putative identified compounds.

Reply: Thanks, that is a good point. In the present work we did not use the data for non-quantified compounds. In the revised manuscript, the following sentence is now added: “In the present work, only the metabolites quantified by NMR or LC-MS or by both methods were considered.”

Comment: Line 191-192: « These conclusions are in a good agreement with the quantitative metabolomic data present in Table 1. » In my opinion this sentense should be removed because data from table 1 have been used to build PCA score plots.

Reply: We agree, the sentence is removed.

Reviewer 2 Report

The manuscript describes the study of the metabolome in two fish tissues (lens and gills), for two species (with a different trophic level) and in two seasons. A total of 68 metabolites were identified and quantified by the combination of LC-MS and NMR techniques.The analytical method was previously developed for human and rat tissues, and applied in fish.

The manuscript is very-well written and easy to follow, as well as informative. The quantification of metabolites in metabolomics studies is still rare, and as shown in this manuscript, can shed some light in the differences between studied species, tissues and seasonality.

I only have a couple of comments:

1) The authors mentions in several occasions that lens tissues are less active metabolically, and therefore can accumulate metabolic changes. I do not agree or at least fully understand what they mean by that. And I could not find any data in the results to support that statement either. I agree that gills are exposed to the external environment and therefore metabolism may change faster. But the metabolome is implicitly transient, and therefore, any "accumulation" is not possible. Please, expand and add clarification in the manuscript.

2) the authors state that "these conclusions [from PCA analysis] are in a good agreement with the quantitative metabolomic data present in Table 1 (line 191)", but don't really expand on that. How are the concentrations in agreement with the four groups obtained in the scores plot in Figure 3?

3) Why only 9 metabolites were selected for Figure 4? Is it a random number or was any criteria used to determine the # metabolites to compare?

4) Line 207-210 “Only the metabolites which differ significantly (p<0.05, fold change > 1.5) are shown. Numerical values can be found in Supplementary Material (Table S3). The bars expanding to the left from the unity correspond to the elevated level of a metabolite in the gill, and to the right – to the elevated level in the lens". All this information belongs in the figure caption as well. 

5) The authors state that "the metabolomic composition of the fish lens undergoes significant seasonal changes due to the low DO level and low feeding activity in winter" Is this occurring only in the lens? What happens in the gills?

Author Response

Comment 1) The authors mentions in several occasions that lens tissues are less active metabolically, and therefore can accumulate metabolic changes. I do not agree or at least fully understand what they mean by that. And I could not find any data in the results to support that statement either. I agree that gills are exposed to the external environment and therefore metabolism may change faster. But the metabolome is implicitly transient, and therefore, any "accumulation" is not possible. Please, expand and add clarification in the manuscript.

Reply: This comment is correct, but only for tissues with the high rate of metabolomic exchange with surrounding tissues. The lens is an anatomically isolated tissue, and the metabolomic exchange between aqueous humor and lens cells (especially cells in the lens nucleus) is very slow. Thus, in such tissues the "accumulation" is possible. We have added the following sentence at line 319: “Although the metabolome is implicitly transient, the intracellular metabolites can stay inside the anatomically isolated tissue (such as the eye lens) for a long time.”

Comment 2) the authors state that "these conclusions [from PCA analysis] are in a good agreement with the quantitative metabolomic data present in Table 1 (line 191)", but don't really expand on that. How are the concentrations in agreement with the four groups obtained in the scores plot in Figure 3?

Reply: Thank you for useful comment. This sentence is now removed from the manuscript.

Comment 3) Why only 9 metabolites were selected for Figure 4? Is it a random number or was any criteria used to determine the # metabolites to compare?

Reply: Being honest, it is indeed a random number which was chosen because 9 panels fitted nicely into 3x3 figure. In our opinion, the demonstration of only 2-3 metabolites would not be sufficient, while the selection of more than 10 metabolites would take too much space. Therefore, we took 9 important metabolites which demonstrate the most significant differences between genera and the most pronounced seasonal variations.

Comment 4) Line 207-210 “Only the metabolites which differ significantly (p<0.05, fold change > 1.5) are shown. Numerical values can be found in Supplementary Material (Table S3). The bars expanding to the left from the unity correspond to the elevated level of a metabolite in the gill, and to the right – to the elevated level in the lens". All this information belongs in the figure caption as well. 

Reply: Thanks, the figure caption is now corrected according to your comment.

Comment 5) The authors state that "the metabolomic composition of the fish lens undergoes significant seasonal changes due to the low DO level and low feeding activity in winter" Is this occurring only in the lens? What happens in the gills?

Reply: That is a good question, but unfortunately we cannot answer it: the fish gills were collected only at winter. One can assume that the metabolomic composition of gills undergoes the seasonal changes as well, but one needs to perform a completely new set of experiments to prove this statement.